# Raman and X-ray Photoelectron Spectroscopic Study of Aqueous Thiol-Capped Ag-Zn-Sn-S Nanocrystals

**DOI:** 10.3390/ma14133593

**Published:** 2021-06-27

**Authors:** Volodymyr Dzhagan, Oleksandr Selyshchev, Yevhenii Havryliuk, Nazar Mazur, Oleksandra Raievska, Oleksandr Stroyuk, Serhiy Kondratenko, Alexander P. Litvinchuk, Mykhailo Ya. Valakh, Dietrich R. T. Zahn

**Affiliations:** 1V. Lashkaryov Institute of Semiconductors Physics, National Academy of Sciences of Ukraine, 03038 Kyiv, Ukraine; dzhagan@isp.kiev.ua (V.D.); yevhenii.havryliuk@physik.tu-chemnitz.de (Y.H.); nazarmazur@isp.kiev.ua (N.M.); valakh@isp.kiev.ua (M.Y.V.); 2Physics Department, Taras Shevchenko National University of Kyiv, 60 Volodymyrs’ka str., 01601 Kyiv, Ukraine; kondr@univ.kiev.ua; 3Semiconductor Physics, Institute of Physics, Chemnitz University of Technology, 09107 Chemnitz, Germany; oleksandr.selyshchev@physik.tu-chemnitz.de (O.S.); oleksandra.raievska@physik.tu-chemnitz.de (O.R.); 4Center for Materials, Architectures, and Integration of Nanomembranes (MAIN), Chemnitz University of Technology, 09107 Chemnitz, Germany; 5L.V. Pysarzhevsky Institute of Physical Chemistry, National Academy of Science of Ukraine, 03028 Kyiv, Ukraine; 6Forschungszentrum Jülich GmbH, Helmholtz-Institut Erlangen Nürnberg für Erneuerbare Energien (HI ERN), 91058 Erlangen, Germany; o.stroyuk@fz-juelich.de; 7Texas Center for Superconductivity and Department of Physics, University of Houston, Houston, TX 77204-5002, USA; litvin@Central.UH.EDU

**Keywords:** colloidal nanocrystals, Ag_2_ZnSnS_4_, phonons, Ag-Zn-S, Ag-Sn-S, non-stoichiometry, secondary phase, XPS, FTIR

## Abstract

The synthesis of (Cu,Ag)-Zn-Sn-S (CAZTS) and Ag-Zn-Sn-S (AZTS) nanocrystals (NCs) by means of “green” chemistry in aqueous solution and their detailed characterization by Raman spectroscopy and several complementary techniques are reported. Through a systematic variation of the nominal composition and quantification of the constituent elements in CAZTS and AZTS NCs by X-ray photoemission spectroscopy (XPS), we identified the vibrational Raman and IR fingerprints of both the main AZTS phase and secondary phases of Ag-Zn-S and Ag-Sn-S compounds. The formation of the secondary phases of Ag-S and Ag-Zn-S cannot be avoided entirely for this type of synthesis. The Ag-Zn-S phase, having its bandgap in near infrared range, is the reason for the non-monotonous dependence of the absorption edge of CAZTS NCs on the Ag content, with a trend to redshift even below the bandgaps of bulk AZTS and CZTS. The work function, electron affinity, and ionization potential of the AZTS NCs are derived using photoelectron spectroscopy measurements.

## 1. Introduction

Thin film solar cells based on Cu_2_ZnSn(S,Se)_4_ (CZTSSe) absorber layers have gained increasing attention due to their suitable absorption spectrum, high absorption coefficient (10^4^–10^5^ cm^−1^), and nontoxic earth-abundant components [1,2]. However, progress with improving cell efficiency has stopped at a value of about 13% since it has not been possible to further increase the open circuit voltage [3]. The latter problem is generally believed to originate from band tails caused by Cu-Zn antisite defects [4]. Partial substitution of Cu for Ag is supposed to be a promising approach to reducing the antisite defect density and thus the band tailing [5,6]. Theoretical calculations demonstrated that due to a substantially larger ionic radius of Ag^+^ (1.14 Å) in comparison with Cu^+^ (0.74 Å) or Zn^2+^ (0.74 Å), the formation energy of the Ag_Zn_ defect (at 0.2 eV above the valence band edge) is much larger than that of the Cu_Zn_ defect (0.12 eV). Many reports indeed showed an improved lattice (cationic) order in (Cu_1−x_Ag_x_)_2_ZnSn(S,Se)_4_ (CAZTS) or Ag_2_ZnSn(S,Se)_4_ (AZTS), as well as a concomitant increase of the photovoltaic device efficiency [5,7,8], although adverse effects of the substitution were also reported [9]. It should be noted that embedding a high Ag concentration into the (intrinsically p-type) CZTS alters the conductivity to n-type [8]. Besides, for AZTS, the stannite structure seems to be more favorable than the kesterite one that is more common for CZTS [10]. Unlike CZTS, AZTS NCs can be fluorescent [11]. Therefore, despite all presumable similarities between CZTS and AZTS, the Cu-for-Ag substitution brings both new opportunities and challenges. 

One of the routes to obtaining a thin absorber layer, especially for flexible photovoltaics (PV), is the synthesis of nanocrystals (NCs) in a colloidal solution, with subsequent deposition of an NC layer on a (flexible) substrate by printing, spin- or spray-coating, or other techniques [1,12]. Due to cost-effective roll-to-roll processes, flexible PV devices have their advantages for their potential applications compared to PV cells on glass [13]. 

Raman spectroscopy has proved to be a generally accepted structural characterization technique for CZTS and many related compounds, including colloidal NCs [14,15,16,17,18,19,20,21,22,23,24,25,26,27]. Besides high sensitivity to secondary (impurity) phases [20,28,29,30,31,32], Raman spectroscopy requires only tiny amounts of the material for analysis (unlike XRD) and no sample preparation (unlike TEM, which is in addition quite challenging for tiny aqueous NCs). Furthermore, it can probe NCs directly in the as-synthesized solutions. For these reasons, Raman spectroscopy permits a broad screening of the synthesis conditions of new NC compounds, particularly by a detailed variation of each of the components in the whole compositional range. Therefore, Raman spectroscopy was chosen in this work as a primary characterization technique of the CAZTS and AZTS NCs synthesized by a method that we recently proved to be successful for obtaining good quality CZTS NCs [33]. 

Most of the Raman studies of CAZTS and AZTS were performed on poly-/microcrystalline films [34,35,36,37,38,39], with only a few reports on colloidal NCs [9,11,40]. There is a noticeable discrepancy in the literature regarding the dependence of the phonon Raman peak position and width on the Ag content in CAZTS, the NC size, or non-stoichiometry [5,39]. With the growing number of reports on the synthesis and applications of colloidal AZTS NCs [6,9,11,33,40,41], the physics and chemistry of defects and structural phases in such NCs need to be explored. 

In this work, a series of CAZTS and AZTS NCs samples were prepared in aqueous solutions under mild conditions and were systematically investigated in detail by resonant Raman spectroscopy, supported by X-ray photoemission spectroscopy (XPS), X-ray diffraction (XRD), IR absorption by phonons, and UV-vis absorption spectroscopies. The experimental results are supported by density functional calculations.

## 2. Experimental Procedure and Calculations

### 2.1. Materials

AgNO_3_, Cu(NO3)_2_, mercaptoacetic acid (MAA), Zn(CH_3_COO_2_), SnCl_2_, Na_2_Sx9H2O, NaOH, and 5 M aqueous NH4OH were supplied by Merck KGaA (Darmstadt, Germany) and used without additional purification. 

### 2.2. Basic AZTS NC Synthesis

The AZTS colloids were prepared from a mixture of mercaptoacetate (MA) complexes of silver, tin, and zinc reacting with sodium sulfide in aqueous alkaline solutions in the presence of ambient air, at 22–24 °C, and under normal pressure. The synthesis procedure is similar to the CZTS synthesis reported earlier [33].

In a typical synthesis, to 3.0 mL deionized (DI) water, 0.3 mL aqueous 0.5 M SnCl2 solution (containing 4 M NaOH), 3.0 mL aqueous 1.0 M MAA solution, 0.1 mL aqueous 5 M NH_4_OH, 0.15 mL aqueous 1.0 M Zn(CH3COO)_2_ solution, and 3.0 mL aqueous 0.1 M AgNO_3_ solution were added dropwise at intense stirring. Afterwards, 0.5 mL aqueous 1.0 M Na_2_S solution was added, and the final mixture was heated at 96–98 °C (in a boiling water bath) in cylindrical glass vials without reflux for 5 min. The as-prepared colloidal AZTS solution was subjected to purification by precipitating NCs with 2-propanol (5.0 mL 2-propanol to 10.0 mL colloidal solution), centrifugation of the water/alcohol mixture for 3 min at 6000 rpm, collecting the precipitate, and redissolving it in 2 mL of DI water.

### 2.3. Variation of the NC Composition and Stoichiometry

Several sample series differing in composition and ratios of the components were prepared to probe the effects of stoichiometry and Ag-to-Cu substitution on optical and vibrational properties, including a series with a varied Ag/Cu ratio (denoted as Ag_x_Cu_1-x_ZnSnS_4_ or (ACZTS NCs)) and AZTS series with a varied silver content (denoted further as Ag_x_ZnSnS_4_ NCs), a varied zinc content (Ag_2_Zn_x_SnS_4_ NCs), a varied tin content (Ag_2_ZnSn_x_S_4_ NCs), and a varied sulfur content (Ag_2_ZnSnS_x_ NCs). In the Ag_x_Cu_1−x_ZnSnS_4_ series, the Ag-to-Cu ratio was varied, keeping the total Ag + Cu amount constant, while in other series, the content of one of the components was varied, maintaining the ratios constant between all other components. Details of preparation procedures for these series can be found in the Supporting Information (SI) file. 

In a separate series of colloids, one or several constituents were not introduced into the system to let possible binary (such as Ag_2_S) or ternary (Ag-Sn-S) phases form for the purpose of comparison. To check possible effects of Ostwald ripening and recrystallization of AZTS NCs, a series of solutions was prepared with different heating periods from 0 to 50 min, maintaining all other parameters constant.

For the spectral studies, the freshly synthesized and purified samples were drop-casted on cleaned substrates, bare Si for Raman, Si or glass for XRD, Au-covered Si for XPS, and placed for drying in a desiccator under dynamic vacuum. 

### 2.4. Characterization

XRD patterns were taken with a Rigaku SmartLab X-ray diffractometer (Tokyo, Japan) equipped with Ni filtered Cu Kα X-ray source. The measurements were performed in θ-2θ geometry with a step of 0.05° 2θ and acquisition speed of 5°/min. The sharp lines from the Si(100) substrate were identified and subtracted.

AFM images were acquired with an AFM 5500 from Keysight (Agilent, Santa Clara, CA, USA). The AFM tip had a radius of 10 nm, and the Si cantilever a resonance frequency of 273 kHz. The samples were prepared by drop-casting of very diluted (to the order of 10^−5^ M in terms of Ag concentration) colloids on a freshly cleaved mica surface which were then dried in a nitrogen stream at room temperature. No ultrasound treatment of the colloidal samples was performed. The NC size distribution charts were plotted based on AFM measurements of the height profiles for several hundred separate QDs. The AFM images were processed using the Gwyddion software (with “plane subtraction” and “aligning rows” filters and zero leveling) (2.58, Czech Metrology Institute, Brno, Czech Republic) and were marked by using the edge-detection grain marking tool of Gwyddion and the distributions of mean height were plotted for each image. 

Raman spectra were excited using 514.7 nm, 532 nm, 638 nm, and 785 nm solid state lasers or the 325 nm He-Cd laser line and were registered at a spectral resolution of about 2 cm^−1^ for visible and 5 cm^−1^ for UV excitation using a LabRam HR800 or Xplora micro-Raman systems (Horiba, Kisshoin, Minami-ku, Kyoto, Japan) equipped with cooled CCD detectors. The incident laser power under the microscope objective (50×) was in the range of 0.1–0.01 mW.

XPS measurements were performed with an ESCALAB 250Xi X-ray Photoelectron Spectrometer Microprobe (Thermo Scientific, Waltham, MA, USA) equipped with a monochromatic Al Kα (hν = 1486.68 eV) X-ray source. A pass energy of 200 eV was used for survey spectra, 40 eV for Auger spectra, and 20 eV for high-resolution core-level spectra (providing a spectral resolution of 0.5 eV). Spectra deconvolution and quantification were performed using the Avantage Data System (Thermo Scientific, v5.9918, Waltham, MA, USA). The linearity of the energy scale was calibrated by the positions of the Fermi edge at 0.00 ± 0.05 eV, Au_4_f_7/2_ at 83.95 eV, Ag_3_d_5/2_ at 368.20 eV, and Cu_2_p_3/2_ at 932.60 eV measured on in situ cleaned metal surfaces. The NCs samples were measured without using a built-in charge compensation system. The spectra were corrected to the C1s sp^3^ peak at 284.8 eV as the common internal standard for binding energy (BE) calibration [42].

UV-vis absorption spectra were registered using a Black Comet CXR-SR UV/vis/NIR spectrometer (StellarNet Inc., Tampa, FL, USA) equipped with miniature deuterium/halogen lamps as an excitation source and 100 μm slits in the range of 220−1100 nm. The spectra were recorded in standard 10.0 mm optical quartz cuvettes.

### 2.5. Lattice Dynamics Calculations

Calculations of the electronic ground state of CZTS and AZTS materials with kesterite structure were performed within generalized gradient approximation using the Perdew–Burke–Ernzerhof local functional [43] as implemented in the CASTEP code [44]. The plane wave basis set cut-off was set to 800 eV. Both materials clearly show insulating behavior and are direct band gap semiconductors. Fixed electron occupancy constraints were imposed on the self-consistent field energy minimization as a prerequisite for using the linear-response scheme in the phonon calculations, treating the atomic displacements as perturbations [45]. Norm-conserving pseudopotentials were used. Before performing calculations, the structures were relaxed while keeping lattice parameters fixed and equal to the experimentally determined ones so that forces on atoms in the equilibrium position did not exceed 2 meV/Å and the residual stress was below 0.01 GPa. A self-consistent field (SCF) tolerance better than 10^−7^ and a phonon SCF threshold of 10^−12^ were imposed. Integration within the Brillouin zone was performed over a 3 × 3 × 4 Monkhorst-Pack grid in reciprocal space. Such an approach was proven to provide reliable lattice dynamics results for quaternary semiconducting materials with various crystallographic structures [46,47].

## 3. Results and Discussion

### 3.1. Ag_x_Cu_1-x_ZnSnS_4_ (ACZTS) NC Series

Based on our recent experience of successfully synthesizing CZTS NCs in water [33], we started a transition from the copper-based kesterite to the silver-based one through a series of mixed Cu_1−x_Ag_x_ZnSnS_4_ (ACZTS) NCs using the same set of synthesis parameters. By tuning the group-I cation content x from Cu to Ag, a series of NCs with the nominal composition Ag_x_Cu_1−x_ZnSnS_4_ was obtained. Their Raman spectra at λ_exc_ = 514.7 nm excitation are shown in Figure 1a. The spectrum of pure CZTS NCs is dominated by the main peak due to an A_1_ mode vibration at about 332 cm^−1^ and corresponding higher order scattering features at 660 cm^−1^ and 990 cm^−1^. This is in agreement with previous reports on CZTS NCs obtained by the same synthesis route and indicates the formation of 3–4 nm kesterite NCs of good crystallinity [33]. Another piece of evidence that the observed phonon Raman spectrum is characteristic of a single quaternary ACZTS structure, rather than a combination of the spectra of ternary and/or binary secondary phases, is the observation of higher order scattering features (Figure 1a). Noteworthy is the fact that the kesterite structure of ACZTS is distinctly preserved despite a pronounced Sn deficit (Figure 1b).

With an increase of the nominal Ag content in the NCs, the main phonon peak becomes broader and weaker, while bands at about 210 and 260 cm^−1^ arise, with comparable intensity to the A mode in the pure AZTS sample (Figure 1a). An additional feature about 160 cm^−1^ can also be distinguished for AZTS samples. 

The actual Ag/Cu ratio in the whole ACZTS NC series was confirmed by XPS (Figure 1b); therefore, the evolution of the Raman spectra should be explained by the increased amount of Ag entering the CZTS NC lattice. Weakening of the main kesterite mode intensity (at 332 cm^−1^) without its noticeable shift and broadening (except for the pure AZTS sample) can indicate a smaller volume of the kesterite phase formed with increased Ag content, while the simultaneous enhancement of new features at about 210 and 260 cm^−1^ can be indication of impurity (secondary) phases being formed. It is most straightforward to consider first the compounds that contain Ag, i.e., Ag-S, Ag-Zn-S, and Ag-Sn-S. Silver sulfide has Raman modes at 200 and 220 cm^−1^ [48], thus providing the lowest frequency feature in our spectra, centered at about 210 cm^−1^, but no mode around 260 cm^−1^. To the best of our knowledge, for Ag-Sn-S only Ag_6_SnS_8_ is known as a stable phase at room temperature and normal pressure [49]. However, the only reported Raman spectrum of this compound is most likely just a combination of Ag_2_S and SnS_2_ modes [49], while for Ag-Zn-S no report on phonon Raman spectra could be found. Therefore, to establish the vibrational Raman fingerprint of the Ag-S, Ag-Zn-S, and Ag-Sn-S compounds, we synthesized and studied Raman spectra of Ag-containing ternary phases and Ag-S, as discussed later in this manuscript.

### 3.2. Heating of the NCs in Solution

In order to check whether the weakening of the main kesterite peak and the features at lower frequencies in the spectrum of AZTS NCs (Figure 1a) can be influenced by post-synthesis thermal treatment, the as-synthesized AZTS NC solution was subjected to heating at 96–98 °C for 5 to 50 min. This led to only a slight narrowing of the Raman peak without a noticeable change of its peak position, even after 50 min treatment (Figure 2a). At the same time, the XRD pattern already changed after 5 min, and after 50 min the reflexes became very sharp, with several new features appearing (Figure 2b). The transformation of the XRD patterns due to the crystallization of initially amorphous (or poorly crystalline) NCs is not likely, because the as-synthesized NCs are already crystalline, according to TEM data (Figure 2b, inset).

The size distributions of AZTS NCs were probed by atomic force microscopy (AFM). In these experiments, the original purified AZTS NC colloids were diluted with DI water by a factor of 200–300, drop-casted on a freshly cleaved mica, and dried in a nitrogen flow. Figure 3a shows a typical AFM image of AZTS NCs synthesized with 5 min heating. The sample contains 3–5 nm NCs along with occasional NC aggregates as large as 20 nm. The NC size distributions were produced using Gwyddion software for 5 different areas and were averaged, resulting in a symmetric distribution with a center at 3 nm (Figure 3b). Additionally, an exemplary image in Appendix A with profiles shows the presence of 3–5 nm AZTS NCs as well as NC aggregates, clearly evidencing clustering of separate NCs. We found that the variation of the thermal treatment duration from 5 to 50 min did not affect the size distribution of AZTS NCs noticeably.

Therefore, the sharpening of the XRD reflexes at about 30, 48, and 55 degrees after heating (Figure 2b) can be understood in terms of an improved ordering in the lattice of the AZTS NCs, while the additional features at 37, 40, 43, and 50 degrees indicate the formation of secondary phases. 

The fairly weak effect of the thermal treatment of the NC solution on the Raman spectra may indicate the fact that the Raman bandwidth is dominated not by the lattice crystallinity but by other factors, which are not determinant for forming the XRD pattern and TEM images, in particular by a short phonon lifetime due to the spatial variation of the nearest neighbor interaction caused by an inherent cationic disorder. It is well reported that the frequency and FWHM of the main Raman peak of a broad range of multinary metal chalcogenides (CZTS, Cu-In-S, Ag-In-Se, Cu-Sn-S, Cu_x_S) are rather insensitive to the sample non-stoichiometry, because it is mainly due to the vibration of sulfur atoms, with cations being immobile [18,24,50,51,52,53,54]. Based on the Raman and XRD data in Figure 1 and Figure 2, we can assume that at a small extent of the Cu-for-Ag substitution (below 15–20%) the kesterite structure is well preserved. At larger Ag content, secondary phase segregation becomes significant, deteriorating the quality of the main (ACZTS) phase. Moderate thermal heating can improve the crystallite structure of the ACZTS NCs as seen by XRD but has only a minor effect on the vibrational (phonon) spectra probed by Raman spectroscopy. The latter is known to be more sensitive than XRD to the arrangement of the cation sublattice of chalcogenide compounds [14,16,27]. 

The results presented above allowed us to conclude that the synthesis conditions adopted from the protocol optimized for CZTS NCs [33] might not be optimal for ACZTS and AZTS NCs. Therefore, we synthesized and investigated several series of AZTS NCs with varying nominal compositions of each of the four constituents. The results are analyzed in the following subsections. 

### 3.3. Variation of Silver Content—Ag_x_ZnSnS_4_ NCs

Figure 4a shows Raman spectra of a series of Ag_x_ZnSnS_4_ with a nominal Ag content (x) between 0 and 30.

The spectrum at x_Ag_ = 0 coincides with the Zn-Sn-S NC spectrum reported by us for the first time recently [29]. With an increase of the nominal Ag content in AZTS NCs, the features at 210 and 260 cm^−1^ increase in intensity with respect to the 336 cm^−1^ one. Note that according to XPS data, with an increase of the nominal Ag content, the real Zn content decreases from 37 to 23% and the Sn content is constant and low (<5%) (Figure 5a). 

### 3.4. Variation of Zinc Content—Ag_2_Zn_x_SnS_4_ NCs

The Raman spectrum of the sample with x_Zn_ = 0 is likely to be characteristic for a certain Sn-poor Ag-Sn-S structure, in particular Ag_6_SnS_8_ [49], because it cannot be attributed to a combination of Ag-S and Sn-S phases, and no other Ag-Sn-S stable phase at room temperature and normal pressure has been reported so far, to the best of our knowledge. The Raman spectrum reported in [49] looks more like a combination of Ag_2_S and SnS_2_ modes. Therefore, the present work is the first that reports a characteristic Raman spectrum of a Ag-Sn-S compound. It should be noted that the Ag-Sn-S phase is relatively stable, based on the Raman spectrum (Figure 4b), because it does not decay into Ag-S- and Sn-S-type Raman spectra. The characteristic spectra of Ag-S, SnS, and SnS_2_ are analyzed for comparison in the Section 3.5 of this manuscript.

The changes in the spectra of the Zn series are most likely not (only) due to an increase of Zn content but (also) due to a concomitant drastic drop of the Ag and Sn contents (Figure 5b). According to XPS data, the sample with a nominal content of x_Zn_ = 15 should contain almost pure ZnS. However, the Zn-S peak in the UV Raman spectra is nearly of the same intensity in the rest of the samples in the series (Appendix A). Therefore, segregation into Ag-S and ZnS is not likely to occur here, and we can attribute the observed Raman spectra to a certain Ag-Zn-S structure, as discussed in the next section.

### 3.5. Variation of Tin Content—Ag_2_ZnSn_x_S_4_ NCs

From the Raman spectra of the Sn series (Figure 6a) it is obvious that the main feature at 340 cm^−1^ gains in intensity with increasing Sn content, while the intensity of the features at about 210 and 260 cm^−1^ remains unchanged. 

Therefore, the latter features can originate from Ag-Zn-S, because the spectrum of ZnS at this λ_exc_ looks different (Figure 7a) and Ag_2_S has either a single peak at 200 cm^−1^ (at red excitation) or peaks at 200 and 220 cm^−1^ (at blue-green excitation) [48]. There has been no Raman spectrum of the Ag-Zn-S compound reported so far to the best of our knowledge. Therefore, it is crucial to identify the inherent spectrum of Ag-Zn-S and distinguish it from the Ag-S one. This will allow the latter two phases to be distinguished in the spectra of AZTS NCs. For this purpose, we measured Raman spectra of Ag-Zn-S and Ag-S NC samples at different λ_exc_ (Appendix A) and detected a pronounced difference between the two spectra at λ_exc_ = 638 nm (Figure 6b). Moreover, the intensity in the Raman spectrum of Ag-Zn-S is stronger than that of the Ag-S NCs at any excitation (Appendix A), providing additional proof that our Ag-Zn-S NCs are not a combination of Ag-S and ZnS phases but possess their own intrinsic electronic and vibrational structure. This conclusion corroborates the different spectral lineshapes of the Ag-Zn-S and Ag-S NCs samples. Furthermore, the spectrum of Ag-Zn-S NCs is not a combination of Ag-S and ZnS spectra because the relatively sharp peak of ZnS NCs is unlikely to account for a broad feature in the Ag-Zn-S spectrum. The spectrum of Ag-Zn-S NCs is most likely a combination of Ag-S vibrations (resulting in the 210 cm^−1^ feature) and Zn-S vibrations (resulting in the 260 cm^−1^ feature) of a joint/alloyed ternary compound/lattice. 

Earlier we observed Zn-S vibrations in the alloyed NCs of other multinary chalcogenides: CuInS_2_-ZnS [21,22,55], Cu-Zn-In-S [55], and Zn-In-S [56]. The spectrum of ZnS NCs at UV excitation (325 nm) reveals a sharp first-order LO phonon peak at 345 cm^−1^ and a relatively strong second-order feature at 790 cm^−1^ (Figure 7b), which is also distinct from the spectrum of Ag-Zn-S NCs at this excitation, showing only a weak first-order LO peak (Figure 7b). Note that the reference Raman spectrum of ZnS NCs shown in Figure 6b and Figure 7 was measured on a sample consisting solely of ZnS NCs of relatively high crystallinity. Comparing the signal-to-noise ratio of the phonon peak in the latter reference spectrum and the Ag-Zn-S spectrum at λ_exc_ = 325 nm, one can hardly expect any noticeable contribution of minor content of pure ZnS phase (concluded from UV Raman) to the spectrum of Ag-Zn-S NCs at visible excitation (Figure 6b).

It should be noted that in the above-discussed series with the varied nominal Sn content, the real Sn content does not vary much and does not exceed 5% in the whole series. In contrast, the Zn content decreases significantly with an increase of the nominal Sn composition (Figure 5c). Therefore, the evolution of the Raman spectra and of the underlying NC structure is most likely determined by the Zn deficit at larger nominal Sn composition. This allows the conditions for AZTS phase formation to be able to compete with the formation of Ag-Zn-S. The variation of real Zn composition in the Sn series explains the similarity of the evolution of the Raman spectra to that of the Zn series (Figure 4b).

### 3.6. Variation of Sulfur Content—Ag_2_ZnSnS_x_ NCs

It was expected that predominantly silver sulfide or silver zinc sulfide would form at low sulfur content, while an increase in the sulfur content would favor the formation of the kesterite phase. In line with these expectations, we observe that at low nominal S content the Ag-Zn-S features dominate the Raman spectra, while the AZTS feature at 336 cm^−1^ builds up with an increase of the nominal S content (Appendix A). This behavior corroborates the chemist’s expectations in the previous paragraph. Furthermore, the increase of the nominal S content is accompanied by an increase of the real Sn content (from 0 up to 10%) and a decrease of the real Ag content (Figure 5d), which is in agreement with enhancing the AZTS feature in the Raman spectra.

### 3.7. Resonance Effects in Raman Spectra of AZTS NCs

Using Raman spectroscopy with different λ_exc_ has proven to be a powerful tool for selectively probing both compositional and structural heterogeneities of small colloidal NCs [20,28,29,30,31,32,57]. In the previous subsections (Figure 6b and Figure 7), we already discussed some of the resonant effects and referred to Raman spectra at different λ_exc_ (Appendix A). Here, we would like to complete this discussion by considering the evolution of a representative AZTS NC spectrum (Figure 8) with λ_exc_ tuned from UV (325 nm) to IR (785 nm) and discuss two additional effects not mentioned above. 

The first effect that can be concluded from Figure 8 is that the relative intensity of the features in the range of Ag-S and Ag-Zn-S modes, i.e., at 220 and 260 cm^−1^, becomes stronger at red and IR excitation. Based on the fact that at visible λ_exc_ Ag-Zn-S has stronger Raman peaks than Ag-S (Figure 6b), and at λ_exc_ = 785 nm there is hardly any Raman feature seen in the Ag-S spectrum (Appendix A), the resonant behavior in Figure 8 can be attributed to the increased contribution of Ag-Zn-S, but not of Ag-S. 

The second effect is that similar to previous reports on other multicomponent sulfides [55,58], Raman spectra excited with the 325 nm laser line selectively detect minor contents of the ZnS phase in AZTS NC samples. It can be distinguished from the spectral feature of AZTS, which occurs in the same frequency range, by observing the second-order (2LO) feature at 790 cm^−1^ for ZnS NCs (Figure 7b).

### 3.8. IR Phonon Spectra

Infrared absorption by phonons is a complementary technique to phonon Raman scattering. It also provides useful structural information on multinary and hetero-NCs [59,60,61,62], although it is still rarely used compared to Raman spectroscopy [57]. IR absorption does not have the advantage of Raman spectroscopy of selective probing different compounds in the NCs. Instead, being a first-order optical process, it profits from higher sensitivity (the probability of the absorption of an IR photon by lattice phonon is orders of magnitude higher than the scattering of a visible light photon by the same phonon). Moreover, the low spectral density of the exciting light in IR spectroscopy excludes any photoinduced processes in the NCs, including photocorrosion or photooxidation. Only a few publications exploring IR absorption spectroscopy by phonons can be found for the entire broad family of I_2_-II-IV-VI_6_ compounds, including no reports on AZTS and only two works on CZTS [63,64]. No reports of the IR phonon spectra of the related Ag-based ternary compounds could be found as well.

Representative IR phonon spectra of several selected NC samples from this work are presented in Figure 9. As in the case of the Raman spectra, one can establish a characteristic spectral lineshape of the AZTS compound, namely a sharp single band at about 360 cm^−1^ and a broad, apparently multicomponent, feature in the range of 150–300 cm^−1^. The former vibration is absent in the samples that did not show the characteristic AZTS peak ≈ 330 cm^−1^ in the Raman spectra. Apparently, the vibrational IR pattern of the Ag-Zn-S structure encompasses a broad spectral range of about 170–280 cm^−1^, similar to the Raman spectra. The spectrum of the Ag-Sn-S NCs (Ag_30_Sn_15_S_25_) is similar to that of the AZTS NCs, except for the much weaker absorption component of about 280–300 cm^−1^ in the former. Therefore, the absorption around 280 cm^−1^ may be related to vibrations strongly involving Zn atoms, while the sharp peak at 360 cm^−1^, according to DFT calculations, invokes Sn-S bonds, and it is absent in Ag-Zn-S. However, more Ag in the lattice enhances this vibration. The latter assumption is supported by the absence of the latter strong feature in the spectrum of CZTS NCs (Figure 9).

### 3.9. Results of Density Functional Theory Calculations

DFT calculations of the phonon spectrum of kesterite AZTS show the main phonon (A_1_ mode) frequency to be 14 cm^−1^ higher than that of the kesterite CZTS (Figure 10).

In the experimental Raman spectra (Figure 10) one can see only a slight upward shift of the main phonon peak with increased Ag content. This result is in agreement with some literature reports [5,65], although a downward shift was reported by others [9,39,66]. What both groups of reports have in common is the small difference in the peak position compared to the CZTS one. This can be explained by the fact that the most intense A_1_ mode peak is due to a symmetric vibration of the anion without involving any cation motion. The modes that involve cation vibration are not resolved in the present work, but in studies where it was observed as a separate peak, it exhibited a continuous shift toward the low energy side with increasing Ag content, well reflecting the effect of the substitution of Cu^+^ by the heavier and bigger Ag^+^ [65]. Consequently, most of the works on ACZTS nano- or microcrystals reported the main Raman peak in the same frequency range as for CZTS ones, namely 332–338 cm^−1^ [36,40].

### 3.10. UV-Vis Absorption

The UV-vis absorption spectra are one of the key characteristics of an absorber material for photovoltaic applications. Even though the optical bandgap of AZTS and CZTS is due to direct interband transitions, the absorption edge of these materials, especially of NCs, is always much “smeared”. The ACZTS NCs studied here also possess the same typical lineshape (Figure 11). The absorption spectra were plotted in (D*hv)^2^ vs. hν coordinates to determine the direct bandgaps from the intercept of the linear part with the hν-axes. Besides the expected transitions, a noticeable absorption was observed in the range hν < E_G_. Such a “tail” originates from the contribution of sub-bandgap states and/or bandgap fluctuation due to chemical composition variations and structural disorder at the nanoscale.

This lineshape is usually explained by an Urbach tail formed by a high density of gap states, which are present even in the samples with high crystallinity in terms of XRD or TEM data [67]. The dominance of the gap/defect states in determining the absorption onset of CZTS and AZTS NC samples may account for the scatter of the literature results and non-monotonous behavior of the bandgap value as a function of AZTS NCs size or Ag content in the CAZTS. In particular, the E_g_ of 1.48–1.65 eV was reported for highly crystalline 5–7 nm AZTS NCs in [11], which is ≈0.5 eV below the bandgap of bulk Ag_2_ZnSnS_4_, 2.0 eV [10]. Note that both the Raman and XRD peaks of the latter NCs were as broad as those of the NCs in the present work, indicating that it is not purely the degree of an “extended” crystallinity of the NCs that governs XRD and Raman peak broadening. The cationic disorder that causes the absorption (Urbach) tail can lead to an apparent red shift of the absorption edge. No significant change or non-monotonous variation of the bandgap value with Ag content in the CAZTS NCs, up to 20% or even more, was reported in other works [10,66,68,69]. For the spectra of ACZTS NCs of the present study, an increase of the absorption in the NIR tail with an increase of the Ag content may also reflect the enhancement of the structural disorder of the ACZTS phase. However, the main reason for the non-monotonous behavior of the absorption edge with Ag content in ACZTS is the interplay of two competing factors: (i) the bandgap increase due to Ag entering the Cu sites in the kesterite lattice of CZTS, and (ii) the red shift of the absorption related with an increasing portion of the Ag-Zn-S phase (as confirmed by Raman spectroscopy, Figure 1), because the absorption edge of pure Ag-Zn-S NCs is at longer wavelengths than that of AZTS or CZTS (Figure 11). These results also reveal the reason why the presumable contribution of the Ag-Zn-S phase to the Raman spectra, i.e., the features at 210 and 260 cm^−1^, gets stronger at red/IR excitation (see Figure 8). 

### 3.11. XPS Study: High-Resolution Core Level, Auger, Valence Band, and Secondary Electron Cut-Off Spectra

In addition to providing NC composition, XPS provides the chemical states of the elements using the high-resolution core level and Auger spectra. A similar study for CZTS NCs was reported previously in [33]. Figure 12 shows the Sn3d, Ag3d, Sn MNN, and Ag MNN photoelectron spectra after correction to the C1s sp^3^ carbon peak at 284.8 eV. Since the Sn3d region directly overlaps with the Zn LMM Auger series, the components are separated by Voigt profiles fitting with a Shirley-type background (Figure 12a). For AZTS NCs, the Sn3d_5/2_ component occurs at the binding energy (434.1 ± 0.1) eV. The kinetic energy of the Auger Sn M_4_N_4,5_N_4,5_(^1^G_4_) feature determined by a derivative method is obtained at (486.2 ± 0.15) eV. The sum of these parameters gives the so-called modified Auger parameter, α’, which is free of a possible charging effect and in the case of Sn is recommended for chemical states analysis. For AZTS NCs, α’_Sn_ = (920.3 ± 0.25) eV supports the attribution to Sn^4+^. For comparison, CZTS NCs reveal α’_Sn_ = (920.2 ± 0.25) eV, for SnS_2_ α’_Sn_ = (920.3 ± 0.25) eV [29], while for oxide SnO_2_ α’_Sn_ = 918.3–919.2 eV [70]. Therefore, in AZTS NCs, XPS confirms the presence of Sn^4+^ in sulfur coordination.

The Ag3d_5/2_ peak at (368.0 ± 0.1) eV, the Auger Ag M_4_N_4,5_N_4,5_(^1^G_4_) feature at (356.7 ± 0.15) eV, and the modified Auger parameter α’_Ag_ = (724.7 ± 0.25) eV support the attribution to Ag^+^_._ According to the literature, Ag_2_S and Ag_2_O reveal the α’_Ag_ at 725.3 eV and 724.5 eV, respectively, while for metallic Ag^0^ α’_Ag_ = 726.12 eV [42], and in our own measurements on a freshly etched polycrystalline silver layer α’_Ag_ = (726.25 ± 0.10) eV. This indicates that there is no reduction to metallic Ag under the mild synthesis conditions used to produce AZTS NCs.

The Zn2p_3/2_ at (1021.70 ± 0.1) eV (Appendix A) matches in binding energy that for CZTS NCs [29] and indicates Zn^2+^ chemical state. The FWHM used for fitting the Zn2p spin-orbit components is different due to the Coster–Kronig effect. The broad feature between the peaks refers to an inelastic background and was fitted using the Tougaard function.

The S2p emission is fitted with two spin-orbit doublets (Appendix A). The first doublet with S2p_3/2_ at 161.35 eV corresponds to S^2−^ sulfur in the inorganic core of the NCs. The second doublet with S2p_3/2_ at 162.35 eV corresponds to sulfur atoms shared between sulfur of mercaptoacetic acid (MAA) ligands and the outer layer of the inorganic sulfide core. Since the peak is shifted to higher binding energy by a value of (1.0 ± 0.2) eV with respect to the S^2−^ peak, it can be formally assigned to S^1−^. The optionally possible third S2p component of a free MAA ligand, not bonded to the NC surface, previously observed for CZTS NCs at ~163.6 eV [29], does not occur for AZTS NCs, which indicates that two cycles of purification are sufficient to remove most amount of unreacted starting reagents and water-soluble byproducts. 

The C1s and O1s spectra originating from the MAA ligand are shown in Appendix A. The observed peaks correspond to –COO– and –C–S– functional groups. The carbon peak at 284.8 eV stems from adventitious carbon.

Information about the work function (WF) of the material is obtained using the secondary photoelectron cut-off edge (SECO) induced by X-ray excitation (Figure 13a). The WF of AZTS NCs is determined to be (4.95 ± 0.1) eV. It is important to note that the value is obtained after correction for the charging effect that can reach 0.3 eV for the given sample. The position of the valence band maximum (VBM) relative to the Fermi level is determined from the valence band spectrum (Figure 13b) measured under X-ray excitation. Even though the resolution of the monochromated X-ray (0.45–0.50 eV) cannot reach the resolution of UV photoemission spectra excited by atomic emission lines, e.g., helium, (0.12–0.15 eV), the X-ray has an advantage of greater escape depth of photoelectrons that allows neglecting a contribution of the MAA ligand and adventitious carbon in the valence band spectrum. After correction for the charging effect, the VBM for AZTS NCs is determined to be at (0.65 ± 0.1) eV. The ionization potential, IP = −(WF + VBM), is determined to be (−5.6 ± 0.1) eV, which is slightly lower than that for CZTS, with literature values of −5.1 to −5.3 eV [71], and according to our measurements on CZTS NCs, (−5.1 ± 0.2) eV. The key point is that since the binding energy and kinetic energy scales are opposite to each other, the ionization potential is independent of charging effects and does not contain systematic errors caused by scale correction. Adding the optical bandgap to the IP gives an estimation of the electron affinity (EA). For AZTS NCs, the EA is estimated to be (−3.6 ± 0.25) eV, which is higher than that for a bulk CZTS (EA −3.75 to −4.0 eV [71]) but lower than the EA according to our data on CZTS NCs (−3.1 ± 0.25) eV. The obtained parameters schematically shown in Figure 13c are fundamental characteristics of the obtained AZTS NCs and can be used for designing electronic devices such as solar cells or help in explaining the photocatalytic properties of AZTS reported previously in the literature [10]. 

## 4. Conclusions

Coupling of the vibrational and optical properties in the resonant Raman scattering phenomenon allows unique information to be unveiled about the structural composition and possible heterogeneity even for ultrasmall and non-stoichiometric NCs of multinary compounds. We established that the structure of the (Cu,Ag)-Zn-Sn-S and Ag-Zn-Sn-S NCs obtained by a facile and scalable aqueous synthesis is generally a combination of the main quaternary (AZTS or ACZTS) and secondary Ag-Zn-S phases. The formation of the latter ternary compound is facilitated by an inherent Sn-deficiency unavoidable for the synthesis in water. Based on specific resonant behavior of the phonon Raman spectra, one may assume that the two phases are not completely isolated from each other but have coupled electronic structures. However, the frequency position of the main kesterite Raman feature at about 336 cm^−1^ is not sensitive to the degree of AZTS NCs non-stoichiometry or amount of Ag-Zn-S phase in them. In the Raman spectra, the AZTS phase can be reliably distinguished from the main impurity phases due to significant differences in the frequency of characteristic vibrational fingerprints of Ag-Zn-S (210 and 260 cm^−1^) and Zn-Sn-S (344 and 380 cm^−1^). The identification of the Ag-Zn-S phase allows non-monotonous behavior of the bandgap of CAZTS NCs with Ag/Cu content to be explained consistently. An alternative explanation of similar anomalies in the UV-vis spectra of the CAZTS system is thus suggested, which was earlier most frequently attributed to the effect of Urbach tail due to structural disorder. The Raman and IR vibrational fingerprints of Ag-Zn-S and Ag-Sn-S, established in this work, along with the detected resonant behavior of the former phase, will facilitate the compositional analysis of AZTS and related complex compounds in the future. The work function, electron affinity, and ionization potential of the AZTS NCs, derived using photoelectron spectroscopy, correlate with the data acquired earlier for CZTS NCs.

## Figures and Tables

**Figure 1 materials-14-03593-f001:**
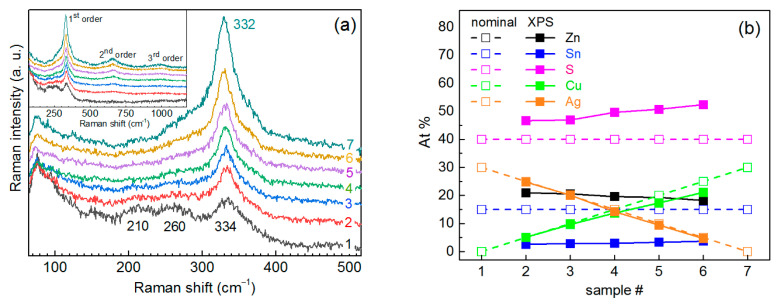
(**a**) Raman spectra of the Ag_x_Cu_1−x_ZnSnS_4_ (ACZTS) NC series (λ_exc_ = 514.7 nm), with the elemental composition of the same samples 1–7 given in (**b**). The inset shows spectra covering the range of 2nd and 3rd overtones of the main peak. (**b**) Nominal (empty symbols, dashed lines) and measured by XPS (solid lines, filled symbols) elemental compositions of the ACZTS NC series.

**Figure 2 materials-14-03593-f002:**
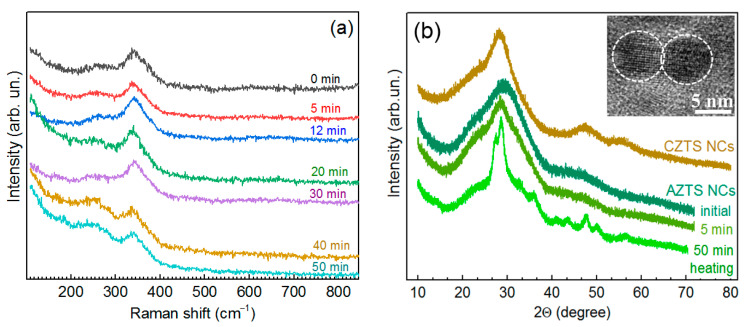
(**a**) Raman spectra of AZTS NCs heated in the solution at 96–98 °C for different times (0 to 50 min). (**b**) XRD pattern of the CZTS and AZTS NCs and AZTS NCs subject to heating in the solution for 5 and 50 min. The inset in (**b**) shows a representative TEM image of as-synthesized AZTS NCs (i.e., not subject to heating).

**Figure 3 materials-14-03593-f003:**
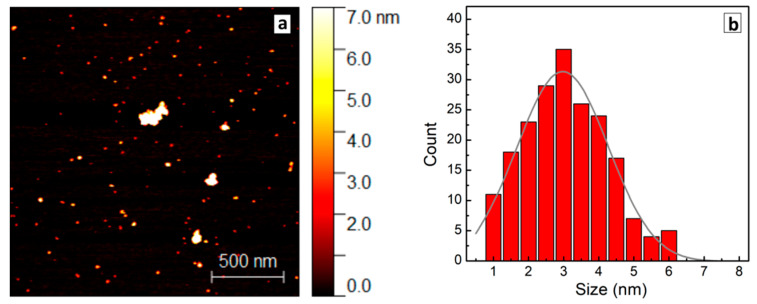
(**a**) AFM image of AZTS NCs on mica. (**b**) Size distribution of AZTS NCs (red bars) derived from the AFM image and fitted with a Gauss profile (solid gray line).

**Figure 4 materials-14-03593-f004:**
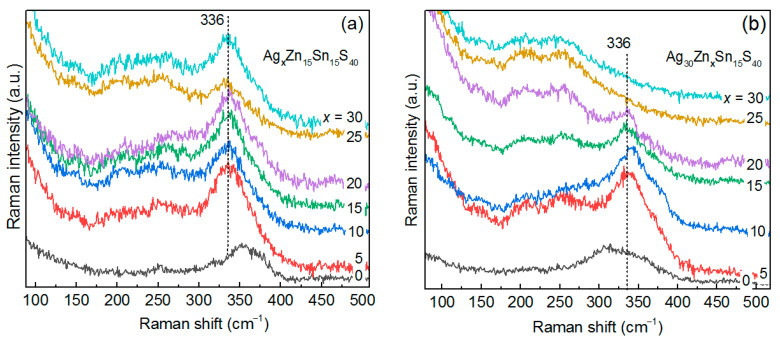
Raman spectra (λ_exc_ = 514.7 nm) of a series of AZTS NCs with a varied nominal content of Ag (**a**) and Zn (**b**).

**Figure 5 materials-14-03593-f005:**
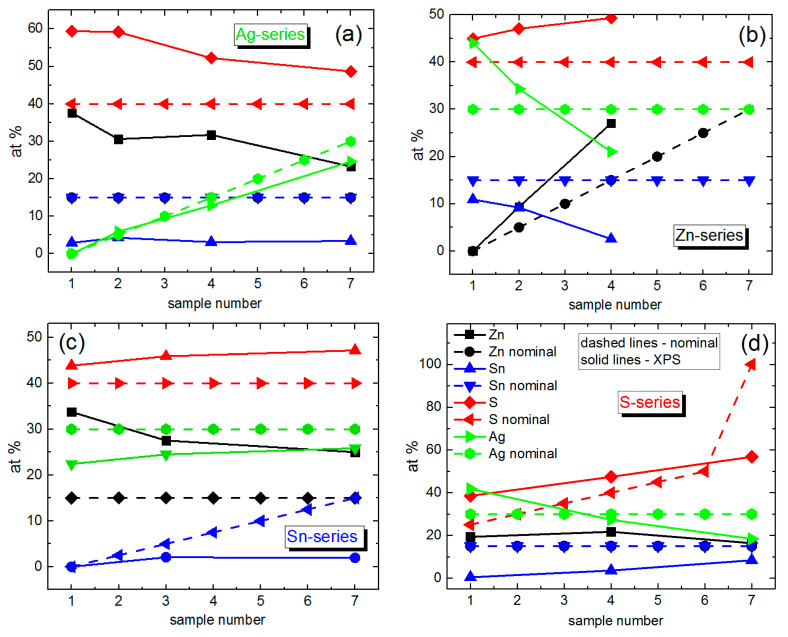
Elemental composition of different series of AZTS NCs with variation of different components: Ag (**a**), Zn (**b**), Sn (**c**), and S (**d**).

**Figure 6 materials-14-03593-f006:**
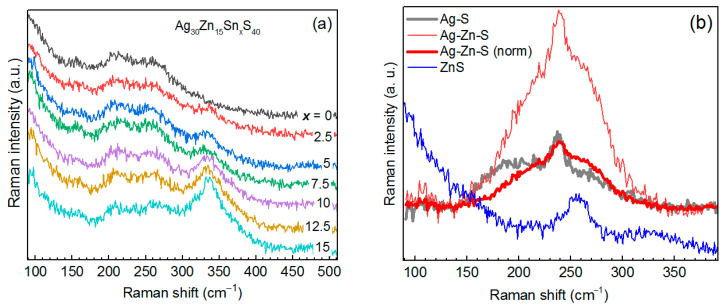
(**a**) Raman spectra of the series of Ag_2_ZnSn_x_S_4_ NCs at λ_exc_ = 514.7 nm. (**b**) Raman spectra of the Ag-Zn-S and Ag-S NCs at λ_exc_ = 638 nm; the Ag-Zn-S NC spectrum is also presented (thick red curve) normalized to that of the Ag-S, in order to underline the different lineshapes for two compounds. The spectrum of ZnS NCs is shown for comparison.

**Figure 7 materials-14-03593-f007:**
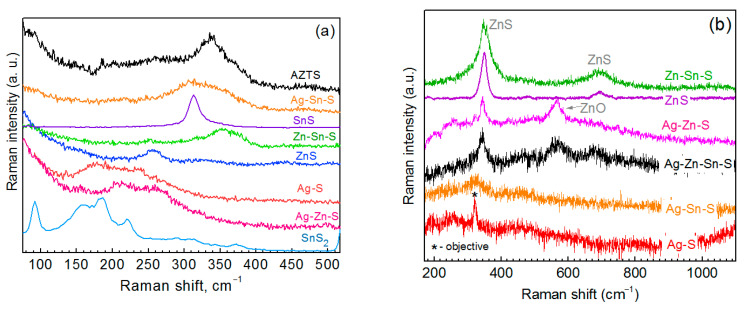
Raman spectra of the typical AZTS NCs and NCs of all possible binary and ternary compounds at λ_exc_ = 532 nm (**a**) and λ_exc_ = 325 nm (**b**).

**Figure 8 materials-14-03593-f008:**
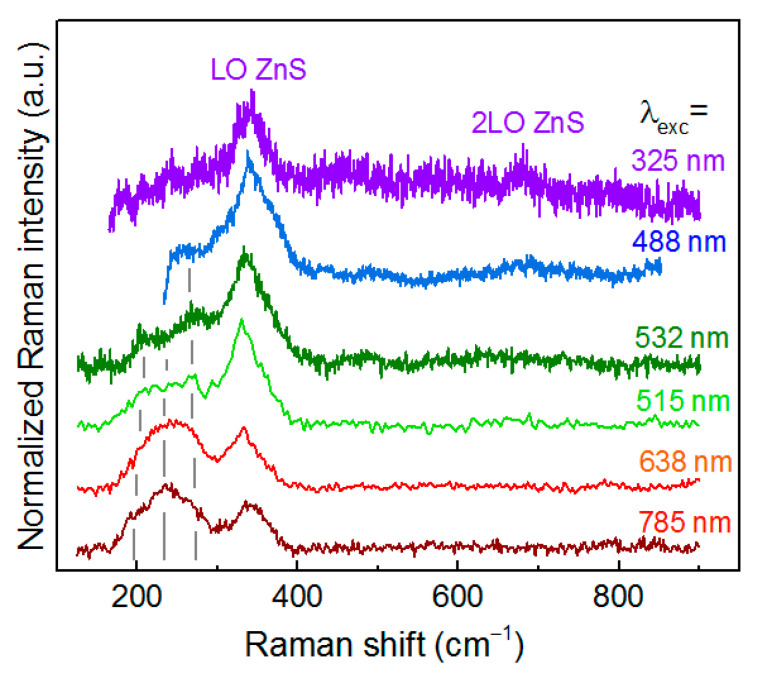
Raman spectra of typical AZTS NCs at different λ_exc_.

**Figure 9 materials-14-03593-f009:**
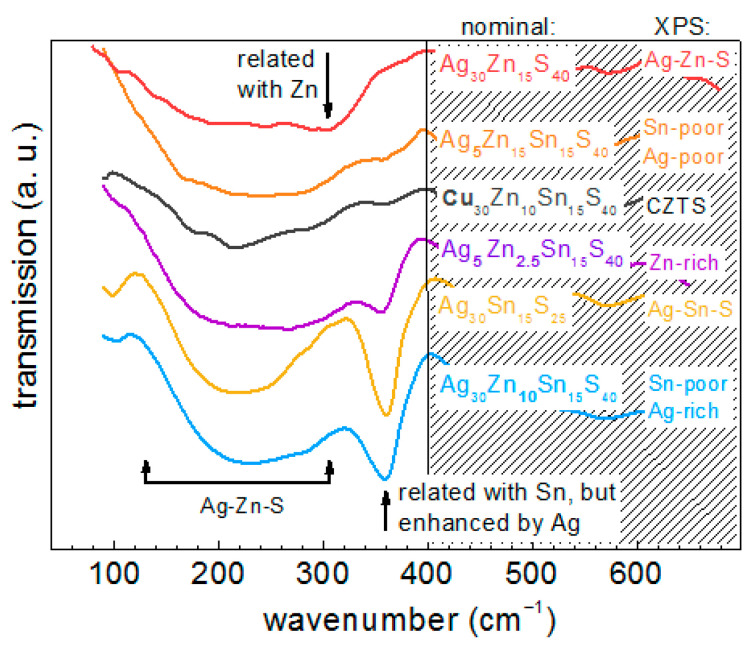
IR phonon spectra of several selected NC samples from this work.

**Figure 10 materials-14-03593-f010:**
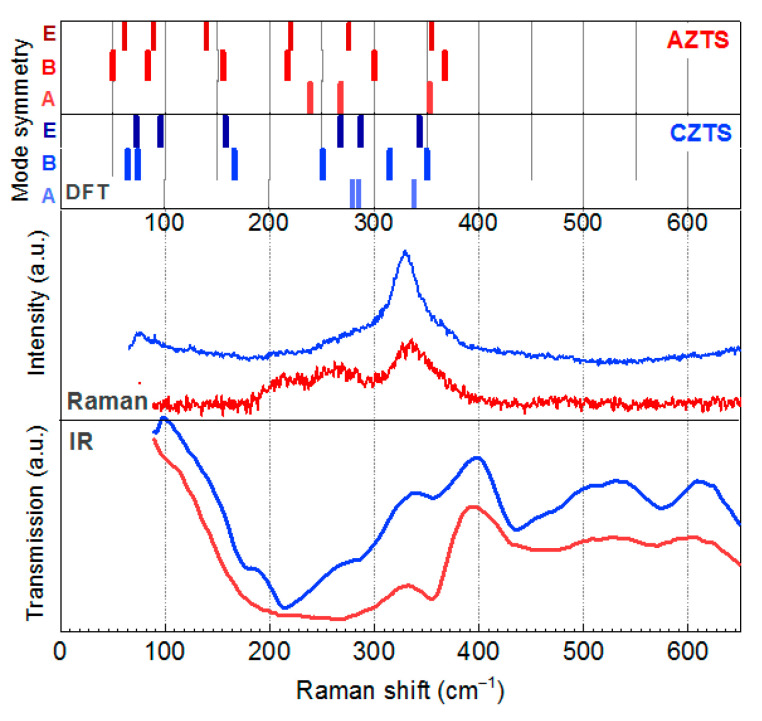
DFT calculated phonon frequencies of AZTS and CZTS kesterite lattices (upper part of the figure) in comparison with the representative experimental Raman and IR spectra of AZTS and CZTS NCs from this work. Note that all modes (A-, B-, and E-symmetry) are Raman-active, while only B- and E-symmetry modes are IR-active.

**Figure 11 materials-14-03593-f011:**
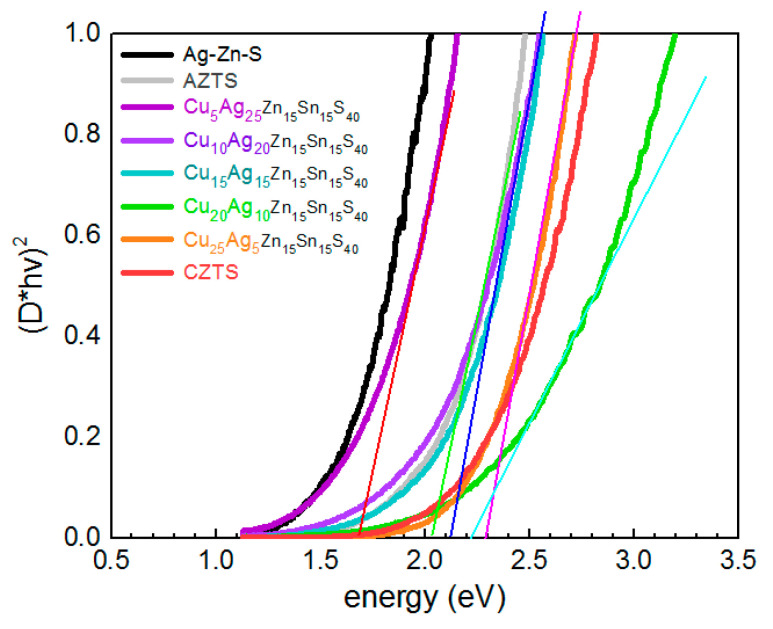
UV-vis absorption spectra of the (Cu_1−x_Ag_x_)_2_ZnSnS_4_ NC series.

**Figure 12 materials-14-03593-f012:**
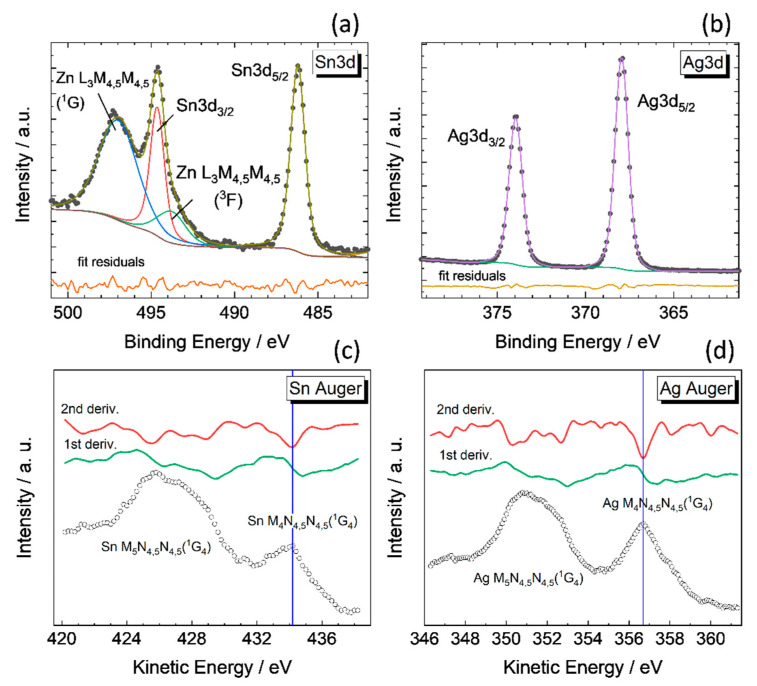
The high-resolution Sn3d (**a**), Ag3d (**b**) core level XPS, as well as the Sn MNN (**c**) and Ag MNN (**d**) Auger spectra of AZTS NCs. The rest of the high-resolution photoemission spectra (S2p, Zn2p, C1s, and O1s) can be found in the Appendix A. The photoelectrons were emitted using a monochromated Al Kα excitation, hν = 1486.7 eV. The sample with the nominal composition Ag_30_Zn_15_Sn_15_S_40_ (XPS determined Ag_30_Zn_22_Sn_3_S_45_) was deposited in the form of a thin film on an ITO substrate.

**Figure 13 materials-14-03593-f013:**
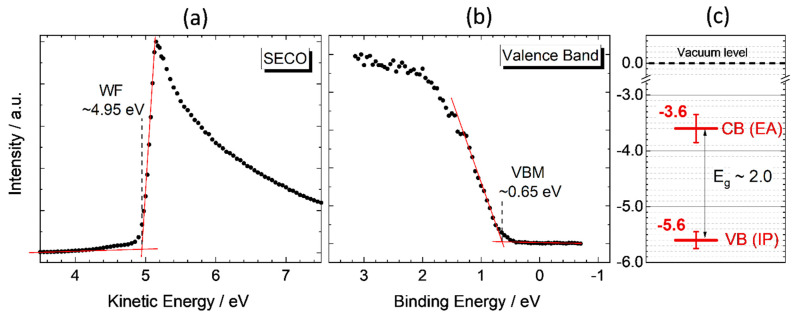
The secondary electron cut-off (SECO) (**a**) and valence band (**b**) X-ray-induced photoelectron spectra. Schematic representation of the valence band (VB) and conduction band (CB) energy levels relative to the vacuum level (**c**). The spectra are obtained under monochromated Al Kα excitation, hν = 1486.7 eV.

## Data Availability

The data presented in this study are available on request from the corresponding author.

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
