# Peer review of "Raman and X-ray Photoelectron Spectroscopic Study of Aqueous Thiol-Capped Ag-Zn-Sn-S Nanocrystals"

_materials, 2021, doi:10.3390/ma14133593_

Round 1
Reviewer 1 Report
Authors of this study reported the synthesis of (Cu,Ag)-Zn-Sn-S (CAZTS) and Ag-Zn-Sn-S (AZTS) nanocrystals (NCs) by means of "green" chemistry in aqueous solution and their detailed characterization by Raman spectroscopy and by several complementary techniques. It was observed that the formation of the secondary phases of Ag-S and Ag-Zn-S cannot be avoided entirely for this type of synthesis. They then demonstrated that the Ag-Zn-S phase that has a bandgap in near infrared range is the reason of the non-monotonous dependence of the absorption edge of CAZTS NCs on the Ag content, with a trend to redshift even below the bandgaps of bulk AZTS and CZTS. Of course, this work is competently done using experimental methods, including analysis of the spectral features of the resulting materials. However, it is not a very well prepared ms. Further revision is necessary, which are given below.
- First four lines of abstract are not necessary, as are too much general. These must be removed, including further improvement of the quality of the abstract.
- Why is DFT calculation detail not given? Why cannot it be appeared just before or after the experimental section? How will the readers understand what the authors did actually? Which software, which functional, which basis set, among other settings, were adopted for such calculations?
- There is a sharp difference between ab initio and DFT! Does the authors of the study understand the underlying difference between them? What did they mean by “Our ab initio DFT calculations …” ??
- Did the authors of the work model the IR frequencies using DFT? Why not the UV-vis spectra, among others?
- Why is the band structure of the systems examined missing in this ms?
Author Response
- First four lines of abstract are not necessary, as are too much general. These must be removed, including further improvement of the quality of the abstract.
Author reply: we removed first two sentences of the abstract, as suggested.
- Why is DFT calculation detail not given? Why cannot it be appeared just before or after the experimental section? How will the readers understand what the authors did actually? Which software, which functional, which basis set, among other settings, were adopted for such calculations?
Author reply: A brief description of the DFT calculation was given in the last paragraph of Section 2 of the manuscript, just before the experimental section (with listed software package, functional etc.). In the revised version of the manuscript this section is expanded.
- There is a sharp difference between ab initio and DFT! Does the authors of the study understand the underlying difference between them? What did they mean by “Our ab initio DFT calculations …” ??
Author reply: Yes, the Reviewer is absolutely correct. Not all DFT is “ab initio”. In order to remove ambiguity in the revised version of the manuscript the “ab initio DFT” phrase is not used. However, as a matter of fact, functionals like LDA or PBE are in fact “ab initio”, as they do not involve empirical parameters and any fitting, unlike, for example, hybrid potentials.
- Did the authors of the work model the IR frequencies using DFT? Why not the UV-vis spectra, among others?
Author reply: In case of non-centrosymmetric kesterite materials (space group I-4, #82) all vibrational modes (A-, B- and E-symmetry) are Raman-active, while B- and E-modes are simultaneously IR-active. In this sense the presented DFT calculations cover IR modes. As such, the upper frame in Figure 10 refers to both Raman and infrared spectra, shown below in the Figure. This fact is specifically mentioned in the Figure 10 caption of the revised version of the manuscript.
As far as UV-vis spectra are concerned: “simple” DFT functionals like LDA or PBE are known not to be appropriate to quantitatively describe the bandgap magnitude of semiconductors. For that reason, we did not attempt to use our DFT results for a bandgap analysis.
- Why is the band structure of the systems examined missing in this ms?
Author reply: There are numerous published papers dealing with the DFT band structure analysis of kesterite CZTS and (more recently) AZTS. For that reason we did not try to “re-invent the wheel” and repeat well documented facts, as lattice excitations were our primary focus.
Reviewer 2 Report
The manuscript titled by “Raman and X-ray Photoelectron Spectroscopic Study of Aqueous Thiol-Capped Ag-Zn-Sn-S Nanocrystals” reports the synthesis of (Cu,Ag)-Zn-Sn-S (CAZTS) and Ag-Zn-Sn-S (AZTS) nanocrystals (NCs) in aqueous solution, and their detailed characterization of structure and composition by Raman spectroscopy and X-ray photoelectron spectrometer (XPS). The optical properties are also measured by inferred IR absorption on phonons and UV-vis absorption spectroscopies. The experimental results are supported by ab initio DFT calculations. The major scientific finding in this work is that the structure of the (Cu,Ag)-Zn-Sn-S and Ag-Zn-Sn-S NCs obtained by a facile and scalable aqueous synthesis is generally a combination of the main quaternary (AZTS or ACZTS) and secondary Ag-Zn-S phases. The formation of the latter ternary compound is facilitated by an inherent Sn-deficiency unavoidable for the synthesis in water. The manuscript is written clearly, and the topic of this work is very interesting due to the potential application in PV industry. However, I cannot recommend the publication of the paper in its current form because of the following issues.
- The paper focuses on the detail data of by Raman spectroscopy and X-ray photoelectron spectrometer. However, the presentation of these data is not well organized. Five groups of NCs are synthesized. It is hard for the reader to follow what is the purpose of these NCs. Do these NCs optimize the method to improve the quality of the NCs?
- XPS only provides the “composition” of these NCs. These data may be useful but not presents well, so the figures (Fig. 1b and 5) of XPS did not bring out any significant new physics. Also it is unclear why the nominal compositions of S and Sn are far away from the XPS data.
- Raman spectra are center of this work. The authors attribute the observed 220, 260 cm-1 Raman spectra to a certain Ag-Zn-S structure. Could the authors confirm this by other means, such as EDS mapping? The existence of Ag-Sn-S suggests that the distribution of Sn is not uniform. Is there a way to improve the quality of the NCs by reducing Sn-deficiency for the synthesis in water?
In a word, the paper did not provide enough new physics and innovation for the field. The presentation of data is also poor.
Author Response
- The paper focuses on the detail data of by Raman spectroscopy and X-ray photoelectron spectrometer. However, the presentation of these data is not well organized. Five groups of NCs are synthesized. It is hard for the reader to follow what is the purpose of these NCs. Do these NCs optimize the method to improve the quality of the NCs?
Author reply: We would like to note that this is a basic approach of any technology of growing multinary compound in general – to vary the nominal composition of the components to obtain material with a desired composition, structure, or other properties. The purpose of studying the series of NCs with the varied composition of each element was finding the nominal composition that gives NCs of best quality in terms of Raman spectra. Our second aim was to study how sensitive/characteristic is Raman (and IR) spectroscopy to different composition of AZTS and to possible ternary and binary compounds that can occur in AZTS as secondary (impurity) phases. For some of these ternary compounds Raman and IR spectra were still missing in the literature, therefore investigating them and reporting to the community was of fundamental interest.
In the text of the manuscript an explanation of studying the composition-varied series of NCs was included on page 6 (bottom paragraph):
" The results presented above allowed us to conclude that the synthesis conditions adopted from the protocol optimized for CZTS NCs [33] might not be optimal for ACZTS and AZTS NCs. Therefore, we synthesized and investigated several series of AZTS NCs with varying nominal compositions of each of the four constituents. The results are analyzed in the following subsections. "
- XPS only provides the “composition” of these NCs. These data may be useful but not presents well, so the figures (Fig. 1b and 5) of XPS did not bring out any significant new physics. Also it is unclear why the nominal compositions of S and Sn are far away from the XPS data.
Author reply: First of all, measurements like XPS, which provide the composition of a material, is indispensable for the investigation of any new material. Therefore, using it in the present study does not need justification, in our opinion. The work is submitted to the journal Materials, which expects more focus on material properties. In the initially submitted manuscript, we provided new physical knowledge regarding the phonon properties via Raman and IR spectra of several compounds (reported for the first time). XPS was used in this work as a characterization technique with known/predictable outcome. Still, the point of the referee is reasonable. To fill the gap of the XPS part, we extended the paper with high-resolution XPS spectra providing information about the chemical states and presented the estimated ionization potential and electron affinity of the AZTS NCs, which represent the novelty in terms of XPS data. The chapter “XPS study: high-resolution core level, Auger, valence band, and secondary electron cut-off spectra” is added to the text of the manuscript, and a figure is added to the supplementary information.
Concerning the Sn deficit, this effect is mostly related to the chemistry of the investigated aqueous system. Under the given highly alkaline conditions required to achieve precursor solubility, only part of the Sn complex precursor reacts with inclusion of Sn4+ in the NC; the rest remains in solution and is washed out during purification. The sulfur content is slightly deviating from the ideal stoichiometry due to the sulfur rich surface of the NCs. In high-resolution XPS S2p spectra, the surface bound state of sulfur atoms shared between the sulfide core and the MAA thio-ligand can be clearly seen.
- Raman spectra are center of this work. The authors attribute the observed 220, 260 cm-1 Raman spectra to a certain Ag-Zn-S structure. Could the authors confirm this by other means, such as EDS mapping? The existence of Ag-Sn-S suggests that the distribution of Sn is not uniform. Is there a way to improve the quality of the NCs by reducing Sn-deficiency for the synthesis in water?
Author reply: The concern and the suggestion of the reviewer is fully understandable. Ideally, in case of compounds with a fixed composition, they can be quite reliably discriminated with EDS, even within a small NC. However, in view of the small size of our NCs and an inherent non-stoichiometry of both the host AZTS NCs and tentative inclusions of Ag-Zn-S or Ag-Sn-S NCs (which also may be of different composition), an EDS map may not provide convincing arguments and qualitatively better understanding of the NC structure than acquired by other techniques. Nevertheless, we a looking for possibility to perform a (presumably quite extended) EDS study in the nearest future.
In a word, the paper did not provide enough new physics and innovation for the field. The presentation of data is also poor.
Author reply: we believe that that with new data in the revised manuscript and explanations of the data that were already present in its initial version, the reviewer's concern will be relieved.
Round 2
Reviewer 1 Report
Authors of this work have indeed considered my comments and have revised their paper. I have no other comments, and thus suggest publication of this interesting work.
Reviewer 2 Report
The authors has responded all my comments. The manuscript now meets the requirements of journal. The presentation of the paper is improved, now I recommend publication of the paper as its current version.